# Submicronic Filtering Media Based on Electrospun Recycled PET Nanofibers: Development, Characterization, and Method to Manufacture Surgical Masks

**DOI:** 10.3390/nano12060925

**Published:** 2022-03-11

**Authors:** Marta Baselga-Lahoz, Cristina Yus, Manuel Arruebo, Víctor Sebastián, Silvia Irusta, Santiago Jiménez

**Affiliations:** 1Institute for Health Research Aragon (IIS Aragón), 50009 Zaragoza, Spain; arruebom@unizar.es (M.A.); victorse@unizar.es (V.S.); sirusta@unizar.es (S.I.); 2Instituto de Nanociencia y Materiales de Aragon (INMA), CSIC—University of Zaragoza, 50009 Zaragoza, Spain; cyargon@gmail.com; 3Department of Chemical Engineering, University of Zaragoza, Campus Río Ebro—I + D Building, 50018 Zaragoza, Spain; 4Networking Research Center on Bioengineering, Biomaterials and Nanomedicine, CIBER-BBN, 28029 Madrid, Spain; 5Laboratorio de Fluidodinámica y Tecnologías de la Combustión (Liftec), CSIC—University of Zaragoza, 50018 Zaragoza, Spain; yago@litec.csic.es

**Keywords:** PET, fibers, face mask, filtration, electrospinning, nanotechnology, COVID-19, aerosol

## Abstract

The disposal of single-use personal protective equipment has brought a notable environmental impact in the context of the COVID-19 pandemic. During these last two years, part of the global research efforts has been focused on preventing contagion using nanotechnology. This work explores the production of filter materials with electrohydrodynamic techniques using recycled polyethylene terephthalate (PET). PET was chosen because it is one of the materials most commonly present in everyday waste (such as in food packaging, bags, or bottles), being the most frequently used thermoplastic polymer in the world. The influence of the electrospinning parameters on the filtering capacity of the resulting fabric was analyzed against both aerosolized submicron particles and microparticulated matter. Finally, we present a new scalable and straightforward method for manufacturing surgical masks by electrospinning and we validate their performance by simulating the standard conditions to which they are subjected to during use. The masks were successfully reprocessed to ensure that the proposed method is able to reduce the environmental impact of disposable face masks.

## 1. Introduction

Regarding the COVID-19 pandemic, the European Environment Agency [1] reported in June 2021 a waste of 170,000 tons of face masks and estimated an additional global warming potential of 2.4–5.7 million tons of equivalent carbon dioxide (CO${}_{2}$). Although the use of protective masks is significantly associated with sociodemographic factors, social behaviors and prevention policies, it has been established in most of the global population [2,3]. The massive use of protective masks will be linked to the epidemiological progress of the pandemic. In the long term, the hypothesis of a procedural change in preventing infectious diseases is considered. The use of a mask will continue to be vital to preventing the spread. Faced with these scenarios, developing new recyclable, and more sustainable materials for application in respiratory protection equipment would be beneficial.

Current technology allows obtaining fiber diameters at the nanometric scale to achieve better filtration performance [4,5,6]. The diameter of the fibers cannot be reduced indefinitely due to the increase in pressure drop that this entails [7]. One of the emerging methods that has shown the best performance for the manufacture of filter materials in a submicron scale is electrospinning, which shows improved performance compared to conventional manufacturing techniques (that is, melt-blown) [8]. Nanofibers offer very high surface-to-volume ratio due to their small diameters and reduced fabric thickness, making possible to generate porous membranes that do not present exceptionally high pressure drops [9]. Numerous studies have been aimed at characterizing the effect of electrospinning parameters (such as polymer concentration, polymer type, flow rate or electric field, among others) concerning the behavior of the resulting membrane in the capture of aerosols [5,10,11,12,13]. One of the starting parameters during the electrospinning of fibers is the choice of the polymeric base material due to divergence in the resulting morphological and physico-chemical aspects of the resulting electrospun mats [10]. Polyethylene terephthalate (PET) has been the material of choice in research works for its easy processability by electrospinning [14]. The use of PET in the food industry has been extended worldwide due to its chemical characteristics, high recyclability and low cost. The wide availability of disposed PET globally makes it ideal for circular economy strategies. Some works have used electrospun PET recycled from plastic bottles to manufacture membranes for cigarette-derived smoke [15], for fluid microfiltration [16] or air nanofiltration [17]. These publications report particular applications of the material and specific electrospinning conditions without offering a broad point of view on the versatility of PET recycled by electrospinning.

Nonwoven filter media consist of a three-dimensional network of randomly placed fibers where the points of contact between them are relatively infrequent and present a relatively high porosity (∼60–90%) [18,19,20]. Their application in gas filtration has been popularly extended thanks to the balance they offer between filtration and pressure drop [18]. The fibers that make up these filters are usually made from cellulose, glass, quartz, and polymeric materials. Their diameter vary within 0.5 and 50 μm [21]. The capture of aerosol particles by filtering depends on the particle’s aerodynamic diameters and is associated with the transport and adherence between the aerosol and the filter medium. The mechanisms capable of carrying out transport are sifting, sedimentation, interception, diffusion, inertial impact, and combined transport mechanisms [22]. When differentiating submicron matter by its size and associated retention mechanisms, we can distinguish between particle sizes >600 nm, 300–600 nm, and <300 nm. The larger ones (>600 nm) are predominantly retained because they are larger than the inter-fiber pores (interception and sedimentation). The intermediate ones (300–600 nm) can pass through the filter pores but have a high probability of colliding with the fibers (inertial impact). In contrast, the finer ones (<300 nm) follow heterogeneous dynamics and are mainly captured by Brownian diffusion [22,23,24]. On the other hand, adhesion mechanisms include supramolecular interactions such as Van der Waals forces, and electrochemical forces, and chemical bonding, although this is more independent of the particle size and depends on its physical-chemical properties [18,25]. Theoretically, many parameters determine the retention efficiency of the filter material to a greater or lesser extent, which may be associated with both the structural characteristics of the filter and its constitutive fibers and the nature of the retained aerosol. It has been well reported that reducing the diameter of the fibers improves filtration since it increases the probability of impact for the aerosols [7,26,27]. For example, a reduction in fiber diameters from 28.2 to 10.9 μm led to an increased filter efficiency on submicron particles from 12.5% to 84% under the same conditions [7]. This same effect was observed in another study with finer fiber filters, where a filtering efficiency of 38% was obtained for 3.25 μm fibers compared to an efficiency of 98.4% for 1.29 μm fibers on average [17]. Depending on the fiber mesh of the filter, they will have a greater or lesser probability of retention [28,29]. Beyond the fiber diameter itself, there are other morphological aspects, such as the presence of fibers with non-circular cross-sections that increases the surface area and increases the retention rate due to inertial impact [30,31,32].

In the present work, we explore in-depth the role of PET in the production by electrospinning of materials to filter air with various applications, with special attention to the production of masks and personal protective equipment against airborne pathogens, such as the SARS-CoV-2. In addition, we present a method for the production of PET-based surgical masks that is easily scalable and transferable to the industry (Figure 1). As face masks are one of the effective measures to slow down the SARS-CoV-2 transmission, the usage of masks has been massively increased. However, the improper disposal of masks is causing a negative impact on the environment and a serious threat to terrestrial and aquatic ecosystems.

## 2. Materials and Methods

### 2.1. Materials

PET nano and microfibers production was carried out using trifluoroacetic acid (≥99.0%, TFA) and dichloromethane (≥99.8%, DCM) as solvents. Both of them were purchased from Sigma-Aldrich (Darmstadt, Germany). Polyethylene terephthalate was obtained from used water bottles having an intrinsic viscosity of 0.7–0.78 dL/g. The bottles were collected from urban waste. The used PET was dissolved in DCM:TFA (3:1) under continuous stirring for 5 to 10 min at different concentrations: 10, 15, 20, and 25 wt.%. The maximum of 25 wt.% was selected according to the limit of solubility. A minimum of 10 wt.% was selected as the lower limit to maximize electrospinning yield.

### 2.2. Electrospun PET Fibers

An Electrospinner Yflow 2.2 D-500 (Electrospinning Machines/R&D Microencapsulation, Malaga, Spain) was used for the electrospinning of the PET micro and nanofibers. During the optimization of the synthesis, different electrospinning parameters were varied: flow rate of the polymeric solution (0.5, 2.5, 5, and 9.9 mL/h), the distance from the needle to the collector (15, 20, and 25 cm) and the voltage selected to stabilize the Taylor cone (15 to 25 kV). The fabric samples were obtained using a moving needle at variable speed on the Y-axis ($V_{y}$ 25 mm/s and 50 mm/s) and constant on the X-axis ($V_{x}$ 2 mm/s) to obtain an homogeneous mat. Aluminum foil was used on a flat collector having electrical conductivity connected to the negative pole during sample collection.

### 2.3. Microscopic Characterization

The characterization of the materials was carried out using an environmental scanning electron microscope (ESEM, Quanta-FEG 250, Zaragoza, Spain). Samples were prepared on carbon tape on an aluminum slide and sputtered with palladium to promote electron conduction. The fiber sizes were obtained from manual measurements with the free software Image-J (v1.52; National Institutes of Health, 2019) for *n* = 100, where *n* is the number of fibers measured.

### 2.4. Retention Efficiency and Pressure Drop Determination

A saline solution (3 wt.% NaCl in distilled water) was used to produce the aerosols using a Topas-ATM226 generator. The generated microdroplets were dragged with a flow of 3000 NL/h and forced to pass through a tubular air dryer based on silica gel to evaporate the water and obtain solid particles. The particles were lugged into a cabin with controlled extraction where the concentration of particles remained constant. In the submicron range (0.01–0.5 μm), the particle size distribution inside the cabin was measured using a SMPS TSI 3936 composed of an electrostatic classifier (DMA TSI 3081) and a condensation particle counter (CPC TSI 3782). The particles were directed at a flow rate of 0.6 L/min towards a tube with an integrated valve. Depending on the desired measurement, the valve could redirect the flow towards the tested filter holder or towards an exhaust tube without a filter. The filter was placed between bronze discs sealed with Teflon tape, with 30 × 20 mm Teflon washers on each side. The exposed filter area was variable (diameter between 4 and 13.3 mm) to adjust the desired flow rate, but we have typically worked with 13.3 mm to reproduce the expected gas velocities in the mask. The measurements lasted 120 s and were made in duplicate.

The determination of the filtration efficiency on micrometric particles (0.5–10 μm) was carried out following the same scheme and conditions as in the previous section but using an Optical Particle Sizer OPS (TSI 3330) at a flow rate of 1.0 L/min (instead of 0.6 L/min of the DMA). The results retrieved from particles below ∼0.5 μm are less reliable in this range using this piece of equipment due to the high ‘dead times’ observed; for this reason SMPS measurements were preferred in this range (Figure 2). Control measurements were carried out between measurements to calculate the relative efficiency (η) according to Equation (Equation 1), where $C_{up}$ stands for concentration upstream and $C_{down}$ stands for concentration downstream. The retention efficiency is expressed in global efficiency as number of particles. The distribution of particle concentration depending on their diameter can be seen in Appendix A.
(1)$$\eta \;=\;100\times \frac{C_{up}-C_{down}}{C_{up}}.\;$$

The determination of the pressure drop was carried out using alcohol tilted columns connected to both sides of the filter sample holder. Measurements were made with a volumetric flow rate of 0.6 L/min.

## 3. Results and Discussion

### 3.1. Fiber Diameter Control

The parameters used during electrospinning were optimized to control the fiber diameters. Firstly, the polymer concentration in the solutions was varied using 10, 15, 20, and 25 wt.%. The viscosity of the solution consisting of PET dissolved in DCM:TFA was varied depending on the polymer concentration. It has been reported that the fiber diameters increase as the electrospun solution viscosity increases [33,34]. The resulting mean diameter was 0.14 ± 0.04 μm; 0.29 ± 0.18 μm; 0.79 ± 0.59 μm and 1.54 ± 0.84 μm when the polymer concentration was 10, 15, 20 and 25 wt.%, respectively (Figure 3).

### 3.2. Contact Angle Measurement

The contact angle was measured to evaluate the hydrophobicity of the electrospun materials. For this, a drop of distilled water was placed on the surface of the fibers and contact angle was measured using the Dataphysics OCA equipment (Dataphysics Instruments GmbH, Filderstadt, Germany) at room temperature. Contact angle measurements were performed by triplicate.

The flow rate of the electrospinning solution is another parameter that directly affects the characteristics of the resulting fibers. High flow rates increase the amount of polymer available for electrospinning, so fibers result in large diameters [35]. Although Zargham et al. [36] reported the influence of flow rates on the distribution of the fiber diameters and they found similar average diameters (± 20 nm) in all cases irrespectively of the flow rates used. In this work, flow rate variations (0.5, 1.0, 2.5, and 5 mL/h) during electrospinning did not appear to significantly alter the resulting fiber diameter or morphology (Figure 4) and the RSD (Relative Standard Deviation) varied from one sample to another without a clear trend. However, a significant presence of beads was observed in the samples having polymer concentrations of 10 and 15 wt.% as the flow rate of the solution decreased. As the flow rate of the solvent is reduced, the solvents have a longer evaporation time from the tip of the needle to the collector, and the fibers are electrospun with fewer imperfections (i.e., lack of beads). In general, when having high flow rates, the probability of bead formation increases [37]. However, in this work, an increase in bead formation has been observed as the flow rate was reduced. Solutions with low concentrations of PET (10–15 wt.%) presented a reduction in the viscosity of the solution compared to those with high concentrations of PET (20–25 wt.%) and beads are more likely to be formed for the solutions with lower viscosity because under those conditions the elasticity of the electrospun solution shows low relaxation time or low extensional viscosity [38,39,40]. The presence of these imperfections can directly affect filtration since it alters the morphology and homogeneity of the fibers. However, Zheng et al. [41] reported an increase in the filtration efficiency in electrospun membranes having beads. This increase in performance was attributed to a reduced pore diameter and increased filter’s packing density caused by the beads. In this work, the geometry of the fibers was not considered a variable of study. Due to the abundant presence of beads in the fibers produced using a polymer concentration of 10 wt.%, the resulting electrospun mats were discarded for the following filtration tests in order to enhance results reproducibility.

### 3.3. Influence of Electrospinning Parameters on Filtration

Table 1 shows the variation in filtration efficiency under different electrospinning conditions. 1 mL of 25 wt.% PET solution was electrospun over an area of 125 × 125 mm, with a distance between the needle and the collector of 20 cm and a voltage difference of ∼△15 kV. Since the liquid flow rate hardly affected the morphology of the fibers, it was decided to maximize this parameter to improve the electrospinning yield, establishing a flow rate of 9.9 mL/h. Initially (Sample 1), the lateral movement was set to 25 mm/s on the Y-axis and 2 mm/s on the X-axis ($v_{x}$). In Sample 2, the Y-axis’s speed ($v_{y}$) was increased to 50 mm/s, maintaining the X-axis speed. The electric field generated between the needle and collector was first set to ∼△15 kV, while Sample 3 was varied to ∼△10 kV. High voltage differences applied between the needle and the collector reduce the diameter of the fibers, although they also generate greater instability during the electrospinning process [42]. In Sample 4, the distance between the collector and the needle was increased, varying from the initial 20 to 25 cm. The distance between the needle and the collector must be large enough to avoid the formation of pores and morphological deformations and small enough to avoid destabilizing the Taylor cone during synthesis [43]. Considering the above, a greater distance is generally associated with reduced diameter fibers [44]. In Sample 5, the flow rate of the solution was reduced from 9.9 to 6 mL/h. Although in this work, it was preferred to work with high flow rates to improve the productivity during the electrospinning process.

These samples were tested to determine their submicron particle capture performance (Figure 5). The fabric that showed the best performance was Sample 5 (low flow rate: 6.0 mL/h), with an average efficiency of 98.3% and a pressure drop of 0.25 mbar. Next, Sample 4 (produced with an increased needle-collector distance in the needle-collector distance) obtained a yield of 95.1% and 0.19 mbar. Samples 1 and 2 (speed increase in Y-axis) showed similar performances (91.7 and 95.2%, respectively), although sample 1 presented a pressure drop of 0.24 mbar and sample 2 of 0.39 mbar. This could be associated with the fact that Sample 2 presented different thicknesses along its Y-axis due to the long distance covered. The trajectory of the electrospinner needle performs a zigzag movement, where the amplitude of the angles depends on the speed in the Y-axis of the needle. The performance of Sample 3 (△kV reduction) was significantly lower than that of the rest of the samples (77.8%; 0.23 mbar). This could be attributed to the fact that the increase in voltage increased the diameter of the fibers and reduced the efficiency in particle retention, as it has been reported in previous works [17,44,45].

### 3.4. Fiber Diameter and Filtration Efficiency

The filtration efficiency is dependent on the fibers diameter. However, the reduction in fiber diameter implies a detrimental increase in pressure drop. In this work, five samples of filter material have been synthesized using variable relative mass quantities of polymer from different concentrations of the parent polymer solutions (15 and 25 wt.% PET) and their combinations, specifically, 1:0, 3:1, and 1:1, and vice versa. The different solutions were electrospun layer by layer one on top of the other. First, the layer of coarse fibers was made. Once the desired amount of polymer had been electrospun, electrospinning continued on top of the previous one using the lowest polymer concentration (i.e., 15 wt.%). The fibers obtained with 15 wt.% showed a fiber diameter of 0.29 ± 0.18 μm, as described in Section 3.1, while the fibers obtained with 25 wt.% PET reached fiber diameters of 1.54 ± 0.84 μm. Experimentally, filtration efficiency increases with the presence of fine fibers (Figure 6a). Superior filtration performance is obtained as the amount of fine fibers increases (15 wt.% PET). In the same way, the pressure drop increases significantly with the reduction of coarse fibers (i.e., the ones obtained using 25 wt.% PET). The filter thickness is also important to achieve the desired filtration performance. Using a composition of 15–25 wt.% PET (3:1), the retention efficiency against submicron particles increases from ∼66% (in the case of the thinnest membrane; ∼85 μm) to more than ∼99% (in the thickest membrane; ∼360 μm). Figure 6b shows how performance and pressure drop variation depending on the density of the fabric. Therefore, this study shows that it is possible to achieve filters with good filtering capacity—pressure drop performance using combined fiber diameters by simultaneously electrospinning polymeric solutions of different concentrations. Depending on the pressure drop and gas flow requirements, it is possible to design filters with high filtration performance. For example, to develop a KN95-type mask, the finest fibers should predominate (in this case, 15 wt.% PET) in a high-density fabric (i.e., 48.5 mg/cm${}^{2}$), always controlling to remain within the pressure drop limits established in the standards.

### 3.5. Homogeneity across the Fabric

One of the problems encountered during the optimization of the fiber production method was the heterogeneity in the filtration efficiency for different samples/regions of a given fabric (Appendix A). This is attributed to the inherent performance of the electrospinning device despite having a moving needle when depositing polymeric mats on large surfaces. To explore the homogeneity of the filter in the same electrospun fabric, samples along one of the diagonals (∼17.5 cm) were cut (identified correlatively from A to J from end to end). The sample was electrospun using the same conditions than in Sample 4 for being the sample with the best pressure drop—filtering efficiency ratio and high electrospinning productivity. The filtration and pressure drop tests showed high homogeneity of the central section: samples D, E, F, and G showed filtration efficiencies of 93.6, 94.3, 93.9, and 93.3%, respectively, with a pressure drop of 0.38 mbar for all of them except for sample G (0.34 mbar). However, the homogeneity was lost as the samples approached to the edges of the electrospun mat. The intermediate samples (B, C, H, and I) revealed a filtration efficiency of 91.7, 89.4, 90.9, and 90.4%, respectively, with a pressure drop of 0.36, 0.32, 0.36, and 0.32 mbar, respectively. End samples (A and J, Appendix A) markedly reduced the overall efficiency of the sample having filtration efficiencies of 81.7 and 81.5% and pressure drops of 0.08 and 0.24 mbar, respectively. The results of this experiment presumed the need to process the sample after electrospinning and select the most effective area. However, by using dynamic collectors (i.e., rotating mandrel collectors) this limitation could be easily overcome [46,47]. However, in this work we wanted to show that using a conventional electrospinning system it is possible to fabricate filtering media that fulfills the standards to be used as personal protection equipment in face masks.

### 3.6. Reproducibility

Atmospheric conditions, such as relative humidity (RH) and ambient temperature (T), may affect fiber formation during electrospinning. The increase in temperature increases the rate of evaporation of the organic solvents used and also decreases the viscosity of the solution. Humidity has a variable effect depending on the physico-chemical properties of the polymer used [48,49]. During one of the syntheses, the measured RH varied from 35 to 50% in a few hours and the temperature, from 20.4 to 23.9 ${}^{{}^{\circ}}$C. A first sample made during this period synthesis reached 93.1% filtration efficiency, while the final sample obtained showed 74.8%. It is thus necessary to control the environmental conditions at the time of electrospinning in order to ensure the uniformity of the samples generated. The electrospinner used in this work did not allow for a control on the temperature and the relative humidity; however, in other electrospinner models, these parameters can be easily controlled. Again, we wanted to show that with a conventional piece of equipment and without having a climate control unit it is possible to fabricate specific filtering media. Then, an evaluation of the reproducibility of the filter fabrics under similar T and HR was carried out. The particle retention efficiency of the obtained fabrics varied between 90.4 and 94.3% and the pressure drop was maintained (0.38 mbar) in the three samples obtained (Figure 7) in a range of RH of 36–42% and a T of 20.7–22.4 ${}^{{}^{\circ}}$C.

### 3.7. Evaluation of the Retention Efficiency over Time

The stability of the filters generated with this technique was studied after 2 and 4 months of storage at ambient conditions. Six samples of filter material of different thicknesses synthesized under the same conditions were used (25 wt.% PET; DCM:TFA (3:1); flow rate of the solution 9.9 mL/h; applied voltage ∼△15 kV; distance from the needle to the collector h∼25 cm; speed on the X-axis: $v_{x}$ = 25 mm/s; speed on the Y-axis $v_{y}$ = 2 mm/s). In the submicron range (Table 2, Figure 8a), losses in retention efficiency between 0.5 and 15.2% were observed after two months, and between 2.9% and 24.8% after four months of storage inside a plastic bag (protected from sunlight) at room conditions. PET degrades under environmental conditions by hydrolysis and photolysis, causing a decrease in crystallinity and mechanical properties, as well as due to an increase in the hydrophilicity of its surface [50,51]. Although PET bottles have a slow rate of degradation depending on temperature and humidity [52], at the nanometric scale, deterioration is accelerated due to the high surface-to-volume ratio of the electrospun nanofibers. Sammon et al. [53] studied 100 nm PET-based films (M${}_{w}$ = 70,000 Da), and estimated a period of degradation by hydrolysis between 4 to 8 days in deionized water at 90 ${}^{{}^{\circ}}$C. In this way, the morphology of the fibers was negatively altered, affecting filtration. In our work, it was observed that, under the appropriate storage conditions (at room temperature and without UV radiation), the filtration was not reduced for particles larger than 400 nm. The results obtained by the OPS (polydisperse size distribution between 0.5 and 10 μm) showed losses in filtration efficiency between ∼2 and ∼13% in all cases. Specifically, the OPS equipment is designed to work with particle sizes greater than 0.5 μm, so the results obtained may not be significant. For aerodynamic diameters between 1 and 10 μm, the efficiency losses were reduced to a range from ∼0.1 to ∼2% (Table 3, Figure 8b). Specifically, considering the EN 14683:2019 [54] standard corresponding to the regulations applicable to surgical masks, the average aerodynamic diameter of the bacteria used in the filtration test is 3 ± 0.3 μm. Against this particle size, the initial tested samples reported a filtration efficiency of 100, 100, 97.0, 100, 95.3 and 97.3%, respectively, while the samples after 4 months storage showed filtration efficiencies as high as 99.8, 98.8, 98.5, 97.8, 98.2 and 98.6%, respectively, which means efficiency losses of less than 2.2% in all cases.

### 3.8. Development of a Surgical Mask

As a result of this work, the method for manufacturing surgical-type masks has been optimized (Figure 9). Usually, these masks are made up of three layers: an intermediate filter sandwiched between an outer and an inner layer of spun-bond, which protects the filter placed in the middle [55]. The mask layers were synthesized using the conditions described in Table 4. Despite having better retention efficiency using low flow rates during the electrospinning process, the filtration efficiency required in surgical mask standards was also obtained using high flow rates. That made us to increase manufacturing productivity and increase the flow rate from 6.0 to 9.9 mL/h. The distance from the needle to the collector was kept at 25 cm for the production of the filter, while it was reduced to 15 cm for the production of the outer and inner layers. This allowed the synthesis of larger fibers, which were visible to the naked eye at the outlet of the electrospinner jet. While the sandwiched filter had an average fiber diameter of 1.24 ± 0.70 μm, the outer and inner layers were composed of larger fibers (3.18 ± 2.63 μm) that were very heterogeneous. The approximate density of the filter was ∼20 mg/cm${}^{2}$ and the thick middle filtering layer was electrospun to a thickness greater than 500 μm.

The fabric presented a retention efficiency greater than 98.2% against particles between 0.5 and 10 μm and 100% against particles of 3 μm (Figure 10a), with a pressure drop of 0.36 mbar. Compared to a commercial surgical mask, this PET-based fabric had better retaining ability for fine and coarse particles. Filter samples show high variability when exposed to submicron particles, even when performed under the same conditions. This is not the case when it comes to coarse particles. In this case, there was no significant variability in retention efficiency.

In the same way, the fabric turned out to be as hydrophobic as the commercial one (Figure 10b). The spun-bond of the commercial surgical masks presented a contact angle of 124.0 ± 4.5${}^{{}^{\circ}}$, while the outer layers of the PET mask prepared in this work had a contact angle of 126.6 ± 6.9${}^{{}^{\circ}}$. The sandwiched filter of a commercial mask showed a contact angle of 124.3 ± 8.0${}^{{}^{\circ}}$, while the PET filter prepared in this work showed a contact angle of 126.4 ± 3.9${}^{{}^{\circ}}$.

Electrospinning is an easily scalable nanotechnological method, mainly due to its versatility for large-scale production [56,57]. The use of multiple nozzles, as seen in the industrial fiber production, has been investigated for upscaling [58,59]. Electrospun polyacrylonitrile-based masks [13,60] and filter media fabrication [61] are clear examples of large-scale production. Compared to the traditional processes for producing non-woven layers (Table 5), electrospinning offers nowadays lower production capacity. For example, production rates of up to 150 kg/h/m have been reported in meltblown processes (fibers 1–25 μm) using 12,000 nozzles per meter [62]. However, a rate of 30 kg/h/m is obtained in the production of surgical masks by extrapolating the electrospinning conditions used in this work. Additionally, this value might be underestimated because the method must be optimized on an industrial scale. Also, it is necessary to consider that although most of the solvents are evaporated during the electrospinning of the fibers, the fabric samples must be dried to facilitate their extraction from the aluminum foil where they are collected (Appendix A).

### 3.9. Exploring FFP2/KN95-Type Filters

The method for manufacturing FFP2/KN95-type masks has been explored (Figure 11). However, a fully functional mask model was not made due to the difficulty of obtaining a good facial fit. According to the EN 149:2001+A1:2009 standard [63] (specifically, the test methods section EN 13274-7:2020 [64]), FFP2-type masks must have an efficiency equal to or greater than 95% against NaCl particles between 0.01–1 μm (Stokes mean diameter of ∼0.4 μm). In addition, the pressure drop must be less than 2.4 mbar against flows of 95 L/min.

On the one hand, 25-15 wt.% PET (1:1) solutions achieved a global filtration efficiency (0.01–1 μm) of more than 98% in ∼10 mg/cm${}^{2}$ density filter medium. The pressure drop was 1.26 mbar under similar conditions. Therefore, there is a margin to increase the filtration efficiency and stay within the permitted pressure drop limits. Note that similar filtering performances can be obtained with different parameter combinations. For instance, 15 wt.% PET (∼10 mg/cm${}^{2}$) achieved an efficiency higher than 99.9%, but pressure drop increased up to >3 mbar.

### 3.10. Reprocessing of PET Masks

It is well known that single-use masks substantially impact the environment [65,66]. One of the crucial economical limitations of recycling is the diversity in the materials used in their manufacture. While the mask layers are mainly made of polypropylene, the ear loops are made of polyamide or rubber and the nose adjustment (nose strip) has a metallic component [67]. Hence, any recycling is complex due to the different nature of their components. In addition, there are variations in the use of materials and their final conformation, such as the combination of polypropylene with polyethylene to manufacture mask layers [68]. The separation of the mask components is one of the most significant limitations of recycling. Alternatives have been proposed to alleviate these limitations, including the use of other materials [69,70], sterilization of masks [71,72], and the possibility of recycling mask materials using different technologies [67,73]. However, the cost of masks has fallen as demand has increased [74]. Considering that the price of a surgical face mask is nowadays less than 0.1 USD/unit, its disassembly and pathogen-free guarantee makes its reuse uneconomical to compete in any market. For instance, the cost of sterilization reprocessing has been estimated at approximately 1.7 USD/unit [75], although it would represent a significant advantage for the environment [76]. In short, the economic cost of recycle them is nowadays prohibitive considering the existing technologies.

Consequently, sustainable alternatives are required to reduce environmental impact. Another advantage that this new strategy offers for manufacturing surgical masks is their ease of reprocessing. Our PET-based fabrics can be re-dissolved in DCM:TFA (3:1), following the steps above, and re-electrospun without affecting the properties of the polymer or the ones of the resulting mat. Figure 12a shows the filtration efficiency for submicron particles of a sample of initial material and a sample obtained by re-electrospinning the same material. There is a small dispersion in the filtering efficiency associated with the climatological variation between the days of the syntheses, but even the recycled sample showed superior efficiencies. As seen in Figure 12b, the morphology of the fibers is not affected by the reprocessing of the PET.

One of the main limitations in reprocessing respiratory protection equipment is the potential contamination with pathogens after use. In this work we chose DCM and TFA not only because they are able to dissolve PET but also because their denaturalizing ability together with the acidic character of TFA [77,78] would contribute to denaturalize potential microorganisms present during reprocessing. For instance it has been reported by Byers et al. [79] that DCM at very low concentrations (78 mM) already inhibits the growth of methanotrophic bacteria. In a future work we will evaluate the inactivation ability of the solvents used (TFA:DCM) on common pathogens (bacteria and viruses) potentially present on used face masks based on the protein denaturizing ability and acidic character of this organic mixture.

The concept of the surgical face mask here proposed would be slightly different from the conventional single-use masks currently existing. In order to improve the life cycle of PET masks (Figure 13) we propose that during the heat-sealing process itself, two holes would be placed in the top and bottom layers of the mask. In this way, reusable and interchangeable rubber bands could be re-used without interfering with the recycling process. In addition, optionally, adhesive nose wires could be used and easily removed, although it is not essential since they could be removed after PET dissolution.

In a potential industrial production those organic solvents evaporated during the electrospinning process would also be condensed and recycled. Both solvents once condensed can be easily separated by distillation due to their different boiling points at 1 atm (72.4 ${}^{{}^{\circ}}$C and 39.6 ${}^{{}^{\circ}}$C for TFA and DCM, respectively). Any possible residue would be treated as a hazardous waste according to the legislation.

## 4. Conclusions

In this work, the influence of various electrospinning parameters on the filtration performance of fibers based on recycled PET has been evaluated. The parameters that influenced the filtration efficiency the most were the increase in the distance between the needle and the collector and the reduction in the output flow rate of the polymeric solution, in addition to the polymer concentration influence, which was decisive for the final diameter of the fibers obtained. Using the results reported in this work, it is possible to develop a wide variety of filter media with different applications. For example, one of the strategies used was to combine fibers produced by electrospinning different concentrations of the polymer (i.e., 15 and 25 wt.%), achieving increased efficiency for the samples with a superior concentration of small fibers produced using 15 wt.% polymer concentration. This could be useful for the production of higher efficiency masks (i.e., KN95, KN99 or air filters).

Finally, we report an efficient method for the production of surgical masks based on three layers: two thick fibers (3.18 ± 2.63 μm) and an intermediate one made of finer fibers (1.24 ± 0.70 μm). The two layers of coarse fibers serve as filter protection and have neither pressure drop nor good filtration performance. The intermediate layer acts as a filter and has a filtration capacity greater than 98.2% for particles between 0.5 and 10 μm and 100% compared to the 3 μm particles used in the standard. The determined pressure drop was 0.36 mbar, within the ranges established in the same standard. The electrospinning process for obtaining this mask was optimized to maximize productivity, using the highest polymer concentration (25 wt.%) and the highest solvent output flow (9.9 mL/h). The resulting material had a hydrophobicity comparable to that of a commercial surgical mask. Surgical masks are usually tested against 3 ± 0.3 μm particles (following EN 14683 protocol), so we demonstrated in this work that the retention capacity of our mats would not be affected significantly over time (up to 4 months) in the capture of particles of those sizes. In addition, our electrospun mats could be re-dissolved and reprocessed without losing properties using the same conditions as in the initial samples, so it could represent a new circular economy method that would reduce the environmental impact caused by the pandemic.

## Figures and Tables

**Figure 1 nanomaterials-12-00925-f001:**
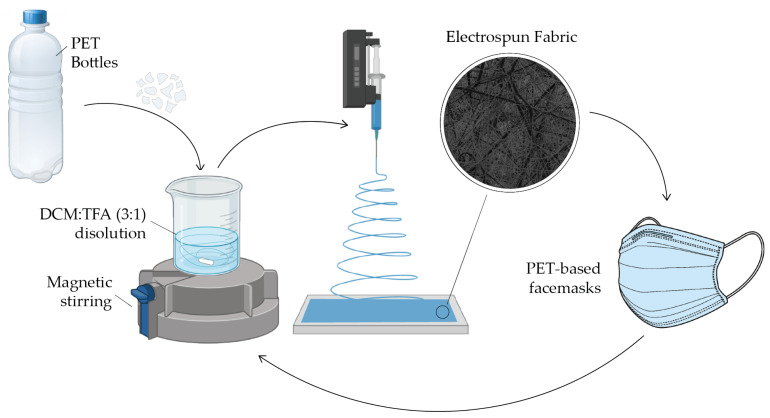
Schematic representation of the PET-based face mask production.

**Figure 2 nanomaterials-12-00925-f002:**
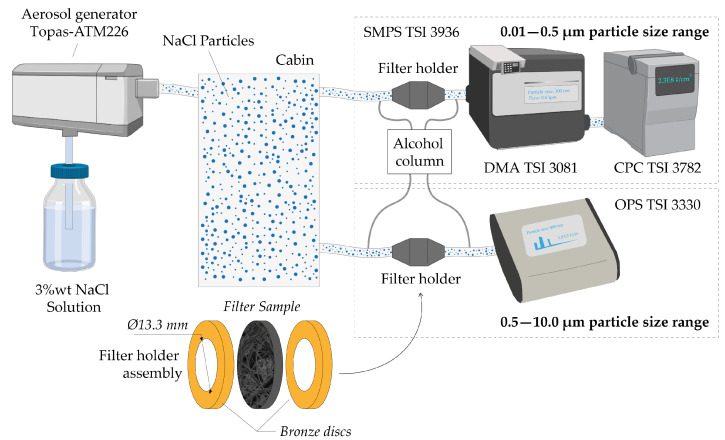
Schematic representation of the equipment used for the particle penetration tests.

**Figure 3 nanomaterials-12-00925-f003:**
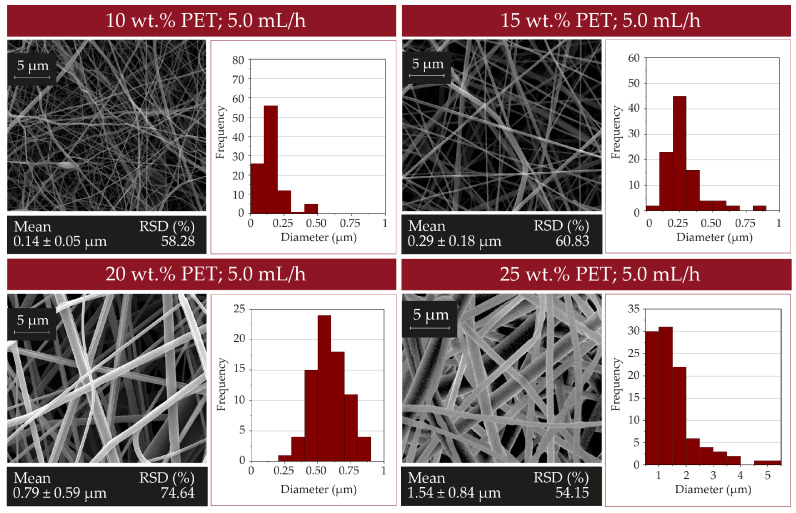
SEM micrographs and fiber size distribution histogram of electrospun fibers (*n* = 100) as a function of the polymer concentration (10, 15, 20 and 25 wt.%) in the solution electrospun at 5.0 mL/h flow rate.

**Figure 4 nanomaterials-12-00925-f004:**
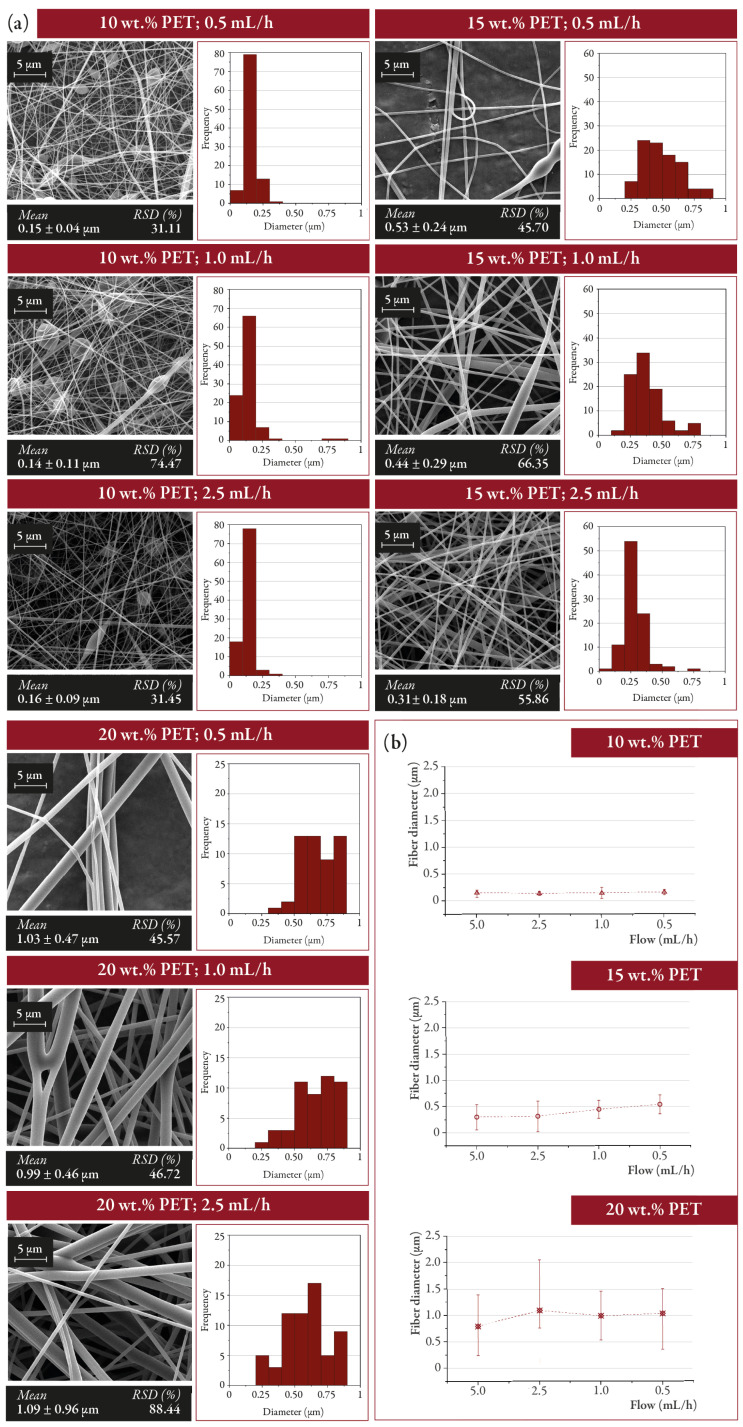
(**a**) Morphology of the electrospun fibers as a function of the polymer concentration (10, 15, and 20 wt.% PET) and the solution output flow (0.5, 1.0, 2.5, and 5.0 mL/h) and (**b**) variation of the diameter of electrospun fibers (*n* = 100) depending on the output flow of the ejected solution.

**Figure 5 nanomaterials-12-00925-f005:**
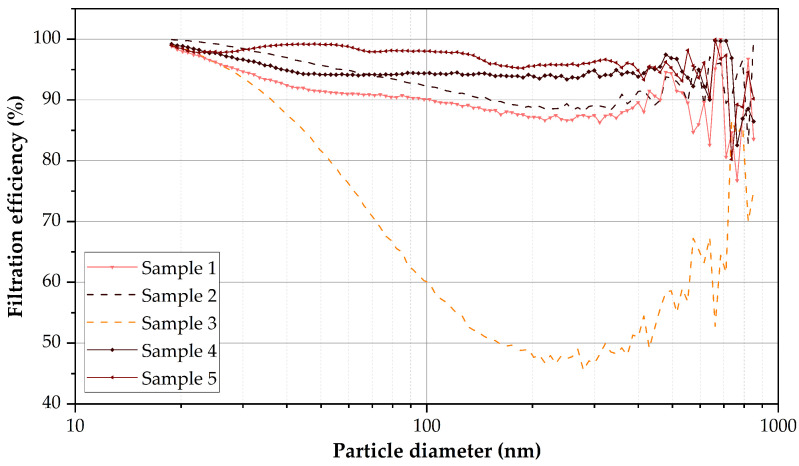
Filtration efficiency using different parameters during electrospinning with a 25 wt.% PET solution. The noise to signal ratio of the curves increases from 0.6 µm because the equipment used is not designed to analyze coarse particles. Particle concentration >0.6 µm (∼1 × 10${}^{5}$ particles/cm${}^{3}$) is reduced compared to that of particles between 0.1 and 0.6 µm (∼2.5 × 10${}^{5}$–7 × 10${}^{5}$ particles/cm${}^{3}$).

**Figure 6 nanomaterials-12-00925-f006:**
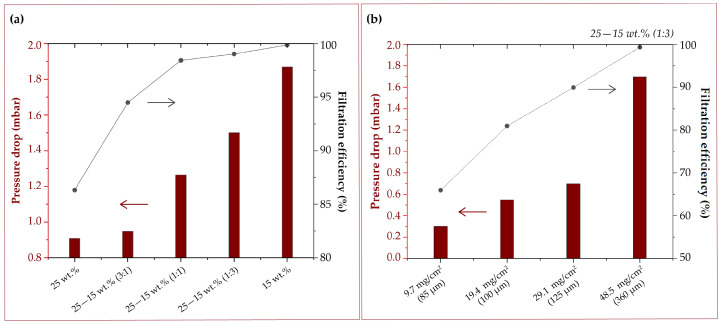
(**a**) Filtering efficiency against submicronic particles (0.01–1.0 μm) and pressure drop against different combinations of fiber diameters obtained by electrospinning solutions of 15 and 25 wt.% PET at different ratios and (**b**) solutions of 15–25 wt.% PET (3:1) at different fabric density.

**Figure 7 nanomaterials-12-00925-f007:**
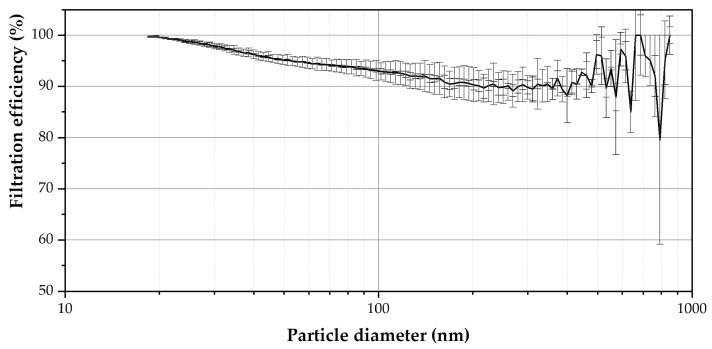
Evaluation of the reproducibility of the electrospinning process at similar atmospheric conditions (RH of 36–42% and a T of 20.7–22.4 ${}^{{}^{\circ}}$C) in triplicate.

**Figure 8 nanomaterials-12-00925-f008:**
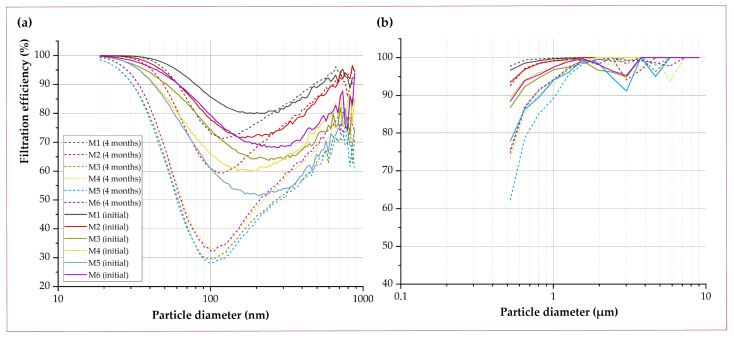
Loss of filtering efficiency after 2 (**a**) and 4 (**b**) months of samples stored at room temperature against 0.01–10.0 μm.

**Figure 9 nanomaterials-12-00925-f009:**
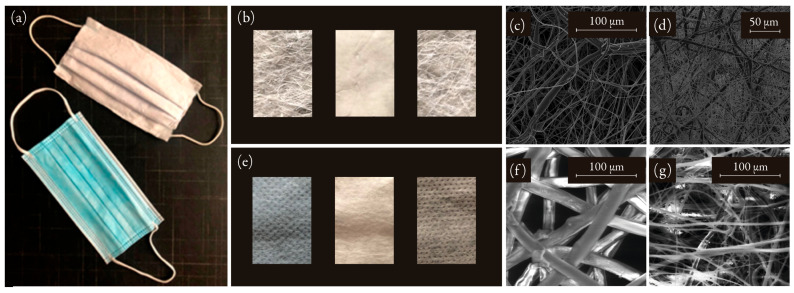
(**a**) PET-based surgical mask developed in this work (top) and a commercial surgical mask (bottom). (**b**) Macroscopic detail of the layers that make up the PET-based mask and (**c**) SEM views of the external layers and (**d**) the filter. (**e**) Macroscopic detail of the layers that make up the commercial surgical mask and (**f**) SEM views of the external layers and (**g**) the filter placed in the middle.

**Figure 10 nanomaterials-12-00925-f010:**
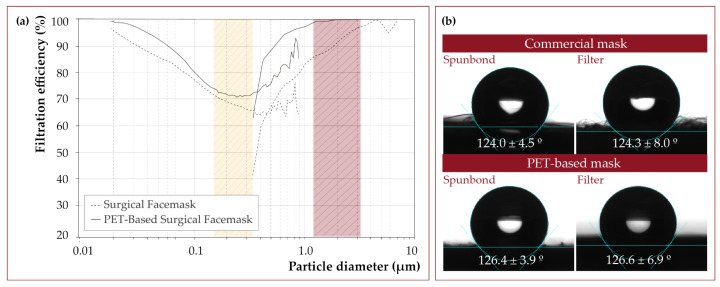
(**a**) Filtration efficiency of a PET-based surgical mask prepared in this work compared to a commercial surgical mask. The polydispersity range of particles used in the EN 14683:2019 surgical mask test is marked in red, and the range used in the EN 149:2001 standard, applicable to FFP2 (similar to KN95) and FFP3 (similar to KN99) masks, is marked in yellow. Labels for filtration efficiency at specific diameters (100, 300, 500, and 700 nm) can be seen on the graph. (**b**) The contact angle of the materials obtained compared to a commercial surgical mask.

**Figure 11 nanomaterials-12-00925-f011:**
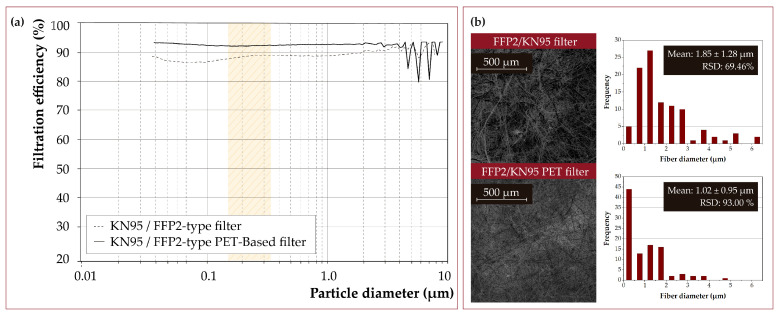
(**a**) Filtration efficiency of a PET-based FFP2/KN95-type filter prepared in this work compared to a commercial filter mask. The polydispersity range of particles used in the EN 149:2001 standard, applicable to FFP2 (similar to KN95) and FFP3 (similar to KN99) masks, is marked in yellow. (**b**) SEM micrographs and fiber size distribution histogram of electrospun and commercial fibers (*n* = 100).

**Figure 12 nanomaterials-12-00925-f012:**
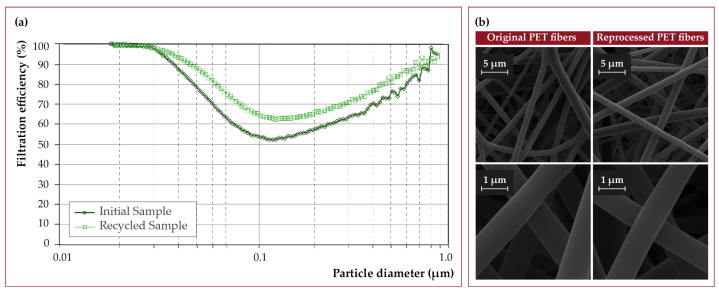
(**a**) Filtration efficiency of an initial PET fabric sample versus recycled (re-electrospun) PET fabric from an equivalent surgical mask and (**b**) SEM views of the PET fibers after reprocessing.

**Figure 13 nanomaterials-12-00925-f013:**
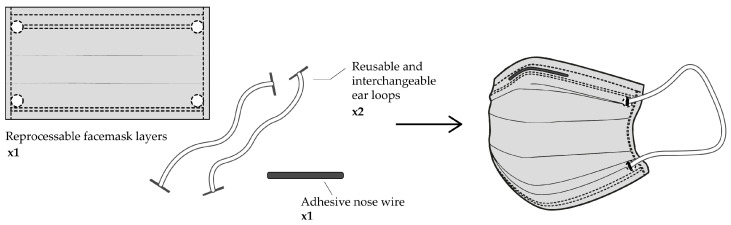
Schematic representation of the PET-based surgical mask concept.

**Table 1 nanomaterials-12-00925-t001:** Variation of parameters used during electrospinning to evaluate their influence on the filtration efficiency of submicron particles.

Sample	PET (wt.%)	Flow (mL/h)	Voltage (kV)	Needle to Collector Distance (cm)	Needle Velocity (mm/s)	Filtration Efficiency (%)	Pressure Drop (mbar)
Sample 1	25	9.9	+10.5/−3.9	20	25($v_{y}$)/2($v_{x}$)	91.7	0.24
Sample 2	25	9.9	+10.5/−3.9	20	50($v_{y}$)/2($v_{x}$)	95.2	0.39
Sample 3	25	9.9	+5.5/−3.9	20	25($v_{y}$)/2($v_{x}$)	77.8	0.23
Sample 4	25	9.9	+10.5/−3.9	25	25($v_{y}$)/2($v_{x}$)	95.1	0.19
Sample 5	25	6.0	+10.5/−3.9	20	25($v_{y}$)/2($v_{x}$)	98.3	0.25

**Table 2 nanomaterials-12-00925-t002:** Loss of filtering efficiency after 4 months of samples stored at room temperature (0.01–1.0 μm particle size range).

Particle Diameter Range 0.01–1.0 μm									
Sample	Thickness (μm)	**Filtration Efficiency (%)** ^1^	Pressure Drop (mbar) ^1^	Filtration Efficiency (%) ^2^	Pressure Drop (mbar) ^2^	Loss of Efficiency (%) ^2^	Filtration Efficiency (%) ^3^	Pressure Drop (mbar) ^3^	Loss of Efficiency (%) ^3^
M1	230	89.9	0.80	87.8	0.86	2.4	87.4	0.88	2.9
M2	210	85.5	0.64	85.1	0.76	0.5	80.9	0.68	5.4
M3	110	78.85	0.40	77.1	0.52	2.2	59.3	0.66	24.8
M4	110	76.6	0.42	65.0	0.36	15.0	63.1	0.70	17.5
M5	100	71.2	0.40	67.7	0.42	4.9	58.3	0.64	18.1
M6	190	82.5	0.46	70.0	0.46	15.2	63.5	0.70	23.0

^1^ Initial values. ^2^ After 2 months. ^3^ After 4 months.

**Table 3 nanomaterials-12-00925-t003:** Loss of filtering efficiency after 4 months of samples stored at room temperature (0.5–1.0 μm and 1.0–0.0 μm).

Particle Diameter Range 0.5–1.0 μm Particle Diameter Range 1–10 μm							
Sample	Thickness (μm)	Filtration Efficiency (%) ^1^	Filtration Efficiency (%) ^2^	Loss of Efficiency (%) ^2^	Filtration Efficiency (%) ^1^	Filtration Efficiency (%) ^2^	Loss of Efficiency (%) ^2^
M1	230	98.1	96.3	1.7	99.9	99.8	0.1
M2	210	96.5	92.5	4.0	99.9	99.8	1.1
M3	110	93.1	86.1	7.0	98.3	98.5	0.0
M4	110	95.2	82.4	12.8	99.7	97.8	1.9
M5	100	89.5	85.5	4.0	97.2	98.2	0.0
M6	190	93.9	82.5	8.5	98.8	98.6	0.2

^1^ Initial values. ^2^ After 4 months.

**Table 4 nanomaterials-12-00925-t004:** Parameters used for electrospinning a surgical mask.

Parameters	Filter	External Layers
PET concentration	25 wt.%	25 wt.%
Voltage	∼△15 kV	∼△25 kV
Needle distance	25 cm	15 cm
Solution flow rate	9.9 mL/h	9.9 mL/h

**Table 5 nanomaterials-12-00925-t005:** Advantages and disadvantages of the electrospinning process.

	Electrospinning Process	Traditional Meltblown/Spunbond Process
**Advantages**		
	Single production equipment can manufacture the outer layers and the mask’s filter.	Two different types of equipment are required to manufacture the meltblown (filter), and the spunbond (external layers) since the change in fiber diameter requires different techniques.
	Electrospun fabrics do not require heat-sealing processes to ensure the bonding of the fibers in the outer layers, although they do require fusing one layer with another.	The spunbond material requires a heat-sealing process to bond the fibers obtained and to form the fabric.
	Electrospinning parameters can be easily manipulated to obtain different fabrics with variable filtration performance.	It is more expensive and complex to manipulate the parameters to obtain different types of filter fabric.
	It offers versatility to functionalize fabrics (e.g., incorporating antimicrobial materials).	The versatility of the process is limited.
**Disadvantages**		
	Toxic solvents are used that must be handled in controlled environments.	It is not necessary to have controlled environments for handling the materials.
	Reduced production rate.	High production rate.

## Data Availability

Not applicable.

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
