# Peer review of "Submicronic Filtering Media Based on Electrospun Recycled PET Nanofibers: Development, Characterization, and Method to Manufacture Surgical Masks"

_nanomaterials, 2022, doi:10.3390/nano12060925_

Round 1
Reviewer 1 Report
The work of Baselga-Lahoz et al. titled “Submicronic Filtering Media Based on Electrospun Recycled PET Nanofibers: Development, Characterization, and Method to Manufacture Surgical Masks” explores the perfomance of recycled PET nanofibers by electrospinning in face masks for personal protective equipment against airborne pathogens, such as the SARS-CoV-2. The realisation of recycled electrospun nanofibers is not new, it has been done by several groups in the past for a wide range of applications. However, understanding and proposing recycled PET nanofibers in facemasks is interesting and I believe solves the key current problem of circularity and recyclability (reuse) of facemasks. Therefore I strongly recommend this work for publication in this journal. However, the critical questions below need addressing
- Since facemask are or should be used for protection, we expect in most cases to have pathogens, viruses and bacteria after use. Comment on how to effectively ensure these are cleaned off the masks during recycling for new products. And at a low cost
- Can the authors comment on the angular flow of particles towards the mask and how this affects the efficiency of filtration? I understand filtration was measured in relatively direct angles between mask and aerosols. What happens if the direction is changed? Does thickness play a critical role here?
- Also, on Page 2 lines 57-58, you attempted to outline the mechanism on which the masks works, however not clearly described. Please use refs https://doi.org/10.1007/s42765-020-00049-5, https://doi.org/10.3390/membranes11040250, https://doi.org/10.1016/j.cocis.2021.101417 to discuss
Author Response
Reviewer: The work of Baselga-Lahoz et al. titled “Submicronic Filtering Media Based on Electrospun Recycled PET Nanofibers: Development, Characterization, and Method to Manufacture Surgical Masks” explores the perfomance of recycled PET nanofibers by electrospinning in face masks for personal protective equipment against airborne pathogens, such as the SARS-CoV-2. The realisation of recycled electrospun nanofibers is not new, it has been done by several groups in the past for a wide range of applications. However, understanding and proposing recycled PET nanofibers in facemasks is interesting and I believe solves the key current problem of circularity and recyclability (reuse) of facemasks. Therefore I strongly recommend this work for publication in this journal. However, the critical questions below need addressing
Answer: Thank you very much for your recommendation and for highlighting the importance of the circularity and recyclability (reuse) of facemasks and the consequent interest of our work.
#1
Reviewer: Since facemask are or should be used for protection, we expect in most cases to have pathogens, viruses and bacteria after use. Comment on how to effectively ensure these are cleaned off the masks during recycling for new products. And at a low cost
Answer: Thank you for your wise query. We specifically chose the mixture trifluoroacetic acid and dichloromethane because not only perfectly dissolves PET but also because bacteria, virus, fungi and protozoa are rapidly inactivated in those organics solvents. TFA is a stronger acid than acetic acid, having an acid ionization constant, Ka, that is approximately 34,000 times higher. Therefore their denaturizing ability and acidic character of TFA would contribute to denaturalizing potential microorganisms present. DCM is also a denaturalizing agent (Pharm Dev Technol. 1998; 3(2):269-76) and therefore microorganisms in contact with those organic solvents will be inactivated.
In response we have added the following sentence in lines 405-414 of the revised version of the manuscript: “One of the main limitations in reprocessing respiratory protection equipment is the potential contamination with pathogens after use. In this work we chose DCM and TFA not only because they are able to dissolve PET but also because their denaturalizing ability together with the acidic character of TFA [78,79] would contribute to denaturalize potential microorganisms present during reprocessing. For instance it has been reported by Byers et al. [80] that DCM at very low concentrations (78 mM) already inhibits the growth of methanotrophic bacteria. In a future work we will evaluate the inactivation ability of the solvents used (TFA:DCM) on common pathogens (bacteria and viruses) potentially present on used face masks based on the protein denaturizing ability and acidic character of this organic mixture.”
#2
Reviewer: Can the authors comment on the angular flow of particles towards the mask and how this affects the efficiency of filtration? I understand filtration was measured in relatively direct angles between mask and aerosols. What happens if the direction is changed? Does thickness play a critical role here?
Answer: Thank you for the concern. The filter pressure drop causes the flow to be perpendicular to it regardless of its orientation away from the surface. As far as retention efficiency is concerned, the fibers can be considered as subjected to cross-flow. Also, the applicable EN standards of efficiency determination tests in facemasks (used in this work) place the global flow perpendicular to the tested fabric.
#3
Reviewer: Also, on Page 2 lines 57-58, you attempted to outline the mechanism on which the masks works, however not clearly described. Please use refs https://doi.org/10.1007/s42765-020-00049-5, https://doi.org/10.3390/membranes11040250, https://doi.org/10.1016/j.cocis.2021.101417 to discuss
Answer: Thank you for the suggestions. We have included in the revised version of the manuscript those references to explain better the mechanisms involved. In lines 59-65 we have included the following sentence: “When differentiating submicron matter by its size and associated retention mechanisms, we can distinguish between particle sizes >600 nm, 300-600 nm, and <300 nm. The larger ones (> 600 nm) are predominantly retained because they are larger than the inter-fiber pores (interception and sedimentation). The intermediate ones (300-600 nm) can pass through the filter pores but have a high probability of colliding with the fibers (inertial impact). In contrast, the finer ones (<300 nm) follow heterogeneous dynamics and are mainly captured by Brownian diffusion [22-24].”

Reviewer 2 Report
The Authors present a work where they "explore in-depth the role of PET in the production by electrospinning of materials to filter air with various applications, with special attention to the production of masks and personal protective equipment", and "present a method for the production of PET-based surgical masks that is easily scalable and transferable to the industry".
Despite the work finalised to the first goal is well constructed and the results well presented and discussed, the second part of the work is not based on a demonstrated need. I mean that the authors should first explain way commercial available facial masks are not recyclable.
While on one side, the significance of the study to investigate the effect of the different electrospinning conditions on the filtering fibers morfology and filtering efficiency is clear, on the other the interest for the recyclability, and the scalability to the industrial production schould be demostrated, for instance by including other relevant references and/or presenting a table with the comparison of advantages and disavantages of the traditional process for the production of non-woven (spun-bond) layers v/s electrospinning.
In conclusion, even if to assess that the electrospunning “could represent a new circular economy method that would reduce the environmental impact caused by the pandemic”
Line 73 -there is a question mark near the references [27?
Lines 83-86 - Please delete or refrase the following sentence “The masks designed in this work were reprocessed, assuring the same filtering quality to confirm that the proposed production approach not only is a sustainable upcycling solution for PET wastes, but also it represents a promising procedure to reprocess the mask production and reduce environmental impacts.” because it is pertinent for the Conclusions, but not well written for the Introduction and aim of the research work.
Lines 89, 90 - trifluoroacetic acid (99.0%,TFA) and dichloromethane (99.8%, DCM) were used as solvents: did the Authors consider the environmentl impact of these solvents? Which is the recovery rate of these solvent in the production process?
Author Response
Reviewer: The Authors present a work where they "explore in-depth the role of PET in the production by electrospinning of materials to filter air with various applications, with special attention to the production of masks and personal protective equipment", and "present a method for the production of PET-based surgical masks that is easily scalable and transferable to the industry".
Despite the work finalised to the first goal is well constructed and the results well presented and discussed, the second part of the work is not based on a demonstrated need. I mean that the authors should first explain way commercial available facial masks are not recyclable.
Answer: Thank you very much for the positive evaluation of our work and for your wise suggestions. Commercial surgical face masks are single use. They are not easily recyclable because they might content biological hazards after use and their inactivation could be costly. In addition, their disassembly (their 3 composing layers) is not straightforward and involves labor. Considering that the price of a surgical face mask is nowadays less than 0,1 USD/unit its disassembly, pathogen-free guarantee and further use would considerably increase its cost and it would not be possible to compete in any market. Also the elastic band is composed of other polymers (i.e., a rubber band made of braided polyester and spandex threads) and they contain a metallic or plastic nose-adjustment (nose strip). All in all it is cheaper to single-use them and dispose them than recycle them. In consequence we have added the explanation in lines 378-394.
Reviewer: While on one side, the significance of the study to investigate the effect of the different electrospinning conditions on the filtering fibers morfology and filtering efficiency is clear, on the other the interest for the recyclability, and the scalability to the industrial production schould be demostrated, for instance by including other relevant references and/or presenting a table with the comparison of advantages and disavantages of the traditional process for the production of non-woven (spun-bond) layers v/s electrospinning.
In conclusion, even if to assess that the electrospunning “could represent a new circular economy method that would reduce the environmental impact caused by the pandemic”
Answer: Thanks for the suggestion. we have added the advantages and disadvantages of the electrospinning method in lines 351-361 and Table 5. Currently, some commercial masks are produced using this technology (see references 60, 61) as well as some filters for other applications (62). In the manuscript we have added comparative information, although the industrial scaling of the technique should be optimized.
Reviewer: Line 73 -there is a question mark near the references [27?
Answer: Thank you. We have corrected it in the revised version of the manuscript.
Reviewer: Lines 83-86 - Please delete or refrase the following sentence “The masks designed in this work were reprocessed, assuring the same filtering quality to confirm that the proposed production approach not only is a sustainable upcycling solution for PET wastes, but also it represents a promising procedure to reprocess the mask production and reduce environmental impacts.” because it is pertinent for the Conclusions, but not well written for the Introduction and aim of the research work.
Answer: Thank you for your query. In the revised version of the manuscript we have deleted it form the introduction section as suggested.
Reviewer: Lines 89, 90 - trifluoroacetic acid (99.0%,TFA) and dichloromethane (99.8%, DCM) were used as solvents: did the Authors consider the environmentl impact of these solvents? Which is the recovery rate of these solvent in the production process?
Answer: Thank you for the concern. According to Regulation (EC) No 1272/2008 TFA shows: Acute toxicity, Inhalation (Category 4), H332. Skin corrosion (Sub-category 1A), H314. Serious eye damage (Category 1), H318. Long-term (chronic) aquatic hazard (Category 3), H412. According to Regulation (EC) No 1272/2008 DCM shows Skin irritation (Category 2), H315. Eye irritation (Category 2), H319. Carcinogenicity (Category 2), H351. Specific target organ toxicity - single exposure (Category 3), Central nervous system, H336. Therefore both organic solvents should be treated before their exhaust to the atmosphere. During the electrospinning process those organic solvents evaporate in the flight from the needle to the collector and no remaining solvents are present in the electrospun fibers. All commercial electrospinners have an exhaust connected to a fume hood.
In the revised version of the manuscript we have added the following sentence (line 422 - 426): “In a potential industrial production those organic solvents evaporated during the electrospinning process will be condensed and recycled. Both solvents once condensed can be easily separated by distillation due to their different boiling points at 1 atm (72.4ºC and 39.6ºC for TFA and DCM, respectively). Any possible residue will be delivered to a hazardous waste manager company”.
